# RepCodec: A Speech Representation Codec for Speech Tokenization

## Abstract

With recent rapid growth of large language models (LLMs), discrete speech tokenization has played an important role for injecting speech into LLMs. However, this discretization gives rise to a loss of information, consequently impairing overall performance. To improve the performance of these discrete speech tokens, we present RepCodec, a novel speech representation codec for semantic speech tokenization. In contrast to audio codecs which reconstruct the raw audio, RepCodec learns a vector quantization codebook through reconstructing speech representations from speech encoders like HuBERT or data2vec. Together, the speech encoder, the codec encoder and the vector quantization codebook form a pipeline for converting speech waveforms into semantic tokens. The extensive experiments illustrate that RepCodec, by virtue of its enhanced information retention capacity, significantly outperforms the widely used k-means clustering approach in both speech understanding and generation. Furthermore, this superiority extends across various speech encoders and languages, affirming the robustness of RepCodec. We believe our method can facilitate large language modeling research on speech processing.

## 1 Introduction

The significant achievements of large language models (LLMs) within the field of natural language processing have attracted considerable attention, as evidenced by notable works such as OpenAI (2023); Brown et al. (2020); Radford et al. (2019); Wei et al. (2021); Chowdhery et al. (2022). Bridging the realms of continuous speech and token-based language models necessitates a key technique known as speech tokenization, which discretizes an audio signal into a finite set of tokens. By converting speech into discrete tokens, language models can predict the future semantic content and generate realistic speech with long-term consistency (Nguyen et al., 2022). As a result, a growing body of research has begun to incorporate speech tokenization into the realm of LLM. Noteworthy examples include AudioLM (Borsos et al., 2023), AudioPaLM (Rubenstein et al., 2023), Vall-E (Wang et al., 2023), PolyVoice (Dong et al., 2023), and SpeechGPT (Zhang et al., 2023a).

Discrete speech tokens can be divided into two categories: semantic tokens and acoustic tokens. Acoustic tokens are produced by audio codecs (Zeghidour et al., 2022; Défossez et al., 2022; Zhang et al., 2023b), which aim to reconstruct the original audio signal so that the audio can be perceptually identical to listeners. However, attempting to preserve all information of the audios leads to high bitrates of acoustic tokens. The process not only imposes significant computational demands on the LLMs, but sometimes makes training infeasible with such lengthy sequences. For example, converting a 30-second audio segment into acoustic tokens results in 18,000 tokens. Therefore, current language modeling approaches often require substantial architectural adjustments to accommodate such long sequences (Borsos et al., 2023; Wang et al., 2023).

Semantic tokens, on the other hand, aim at preserving only the semantic information of the audio, which allows much lower bitrates. If the task relies only on the content of the speech (*e.g.* speech recognition / translation), using semantic tokens should be a better choice. At present, k-means clustering on speech representations (Hsu et al., 2021) is the most prevalent technique of extracting semantic tokens. However, this method has two drawbacks. Firstly, it suffers from a loss of semantic information compared to the original speech representations (Lee et al., 2022b; Borsos et al., 2023). Secondly, not all sets of speech representations are suitable for clustering. Rubenstein et al. (2023)

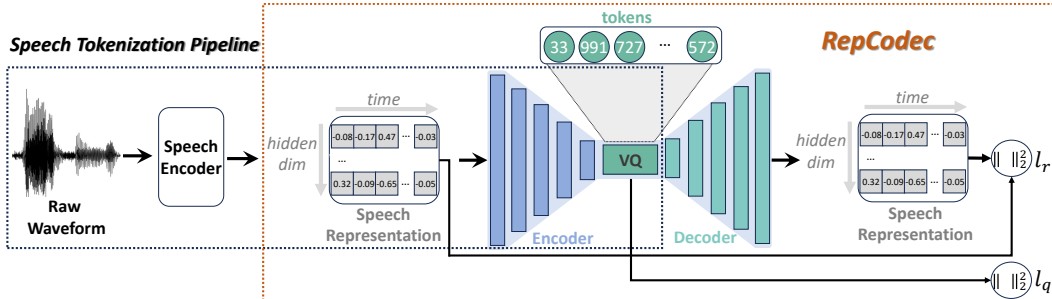

**Figure 1:** RepCodec model architecture. Our network architecture uses single residual units without dimension reduction.

reported that the choice of speech encoder significantly affects the downstream task performance of speech tokenization. Thus, we are motivated to improve the method of extracting semantic tokens by addressing the above problems.

In this paper, we propose RepCodec, a general tokenization approach for representations that can be applied to speech to extract its semantic tokens. RepCodec leverages an end-to-end neural codec to preserve more information of the speech representations. RepCodec is composed of an encoder, a vector quantizer (VQ) and a decoder, and it is trained to reconstruct the input speech representations as much as possible. The speech encoder, codec encoder and the VQ codebook together constitute the speech tokenization pipeline, which can produce high-quality semantic tokens for downstream tasks with a low bitrate. We evaluate the quality of RepCodec tokens in downstream tasks of speech understanding and generation. Specifically, we use a decoder-only automatic speech recognition (ASR) modelling task to evaluate the speech understanding of RepCodec tokens, and a unit-to-speech resynthesis task to measure the quality of RepCodec tokens for speech generation. Our comprehensive experiments demonstrate that RepCodec significantly outperforms the dominant k-means clustering approach. It is worth noting that supervised approach such as ASR and phoneme recognition can also be considered as forms of speech tokenization that convert speech into word tokens or phoneme tokens. However, it is essential to highlight the large amount of parallel data required for supervised training only exists for high-resource languages. RepCodec, on the contrary, is an unsupervised speech tokenization method that can be applied to any languages. The contribution of our work can be summarized as follows:

1. We propose a novel framework, RepCodec, which applies compression techniques on representations to enhance the preservation of information within representations.

2. The experiments show that, by applying RepCodec to speech, semantic tokens exhibit an improved capacity for retaining information, and they surpass the prevailing k-means clustering approach in both speech understanding (4.5% v.s 2.8% word error rate (WER)) and generation (7.6% v.s 4.7% WER). In addition, further experiments demonstrate that RepCodec is a robust method that can be applied to various speech encoders and languages.

3. Our further analysis underscores that the quality of semantic tokens primarily relies on the information loss instead of their similarity to the ground truth phonemes. This finding serves as motivation for future advancements in semantic token refinement.

## 2 RELATED WORK

There are several lines of work related to RepCodec, including self-supervised speech representation learning, speech tokenization and vector quantization.

**Self-supervised Speech Representation Learning.** This line of research has recently gained huge success in the area of speech recognition. Prevalent methods usually requires the model to predict the content of unseen regions (Hsu et al., 2021; Baevski et al., 2022; Chung et al., 2021; Chung & Glass, 2020) or to contrast the target unseen frame with randomly sampled ones (Baevski et al., 2020; Conneau et al., 2021). Specifically, HuBERT (Hsu et al., 2021) is a pioneering work to employ k-means for speech tokenization. The generated tokens serve as the training targets for the speech encoders. Furthermore, they find that these tokens exhibit a strong correlation with the phonemes. Later on, Lee et al. (2022b); Meng et al. (2023) show that these semantic tokens can be used directly

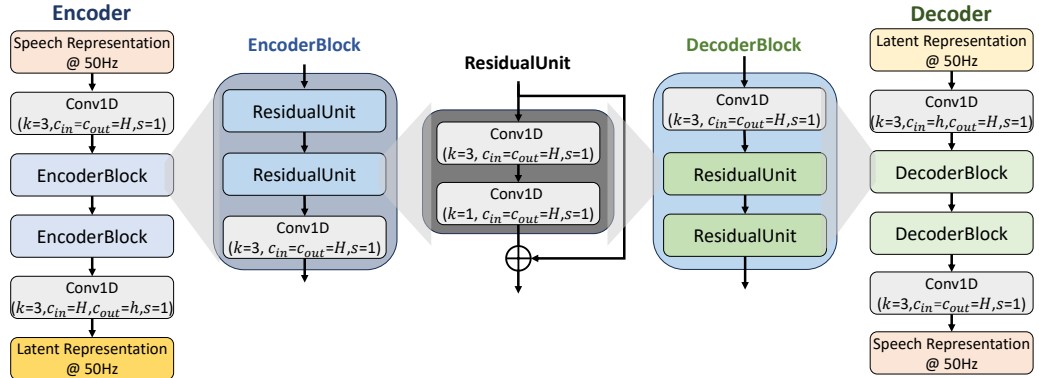

**Figure 2:** Encoder and decoder architecture of RepCodec. $k$ denotes kernel size, $c_{in}$ and $c_{out}$ denote input and output channels, $s$ denotes stride, $h$ denotes the number of clusters, and $H$ denotes the hidden dimension of input representations.

to perform downstream tasks like ASR or speech resynthesis. In addition to the self-supervised approaches that only use unlabeled speech data, many works train the speech encoder in an end-to-end manner (Gulati et al., 2020; Radford et al., 2023) with labeled speech data. Among them, Whisper (Radford et al., 2023) is currently the largest open-source model for speech recognition, which achieves the lowest overall WER across various domains.

**Speech Tokenization.** Discrete speech tokens can be divided into two categories: *Semantic tokens* (Lee et al., 2022b; Hsu et al., 2021; Chung et al., 2021) and *Acoustic tokens* (Zeghidour et al., 2022; Défossez et al., 2022; Wu et al., 2023). Semantic tokens maintain the linguistic information of the speech and have high correlation with phonemes (Hsu et al., 2021). They are commonly generated by applying k-means clustering to pretrained speech encoders like HuBERT (Hsu et al., 2021) or data2vec (Baevski et al., 2022). Semantic tokens are widely used for downstream tasks. For example, Lee et al. (2022b); Polyak et al. (2021); Lee et al. (2022a) use them to train a unit-vocoder to generate raw speech. AudioLM (Borsos et al., 2023) inputs the semantic tokens of w2v-BERT to represent semantic information of the audio. Zhang et al. (2023a) jointly trains a language model with semantic tokens to inject speech recognition ability to GPT-like models. Dong et al. (2023) also incorporates semantic tokens for speech to speech translation. However, the discretization step of k-means discards plenty of information of the speech, resulting in degraded performance in downstream tasks (Lee et al., 2022b; Borsos et al., 2023).

In contrast to semantic tokens, acoustic tokens aim to preserve all the information of the audio. Soundstream (Zeghidour et al., 2022) and EnCodec (Défossez et al., 2022) use a neural audio codec with Residual Vector Quantizers (RVQ) to learn acoustic tokens that can be directly reconstructed into raw audios. As these acoustic tokens contain acoustic information of the audio, they can be used to perform more complicated tasks than semantic tokens. For example, VALL-E (Wang et al., 2023), uses them for zero-shot text to speech (TTS), and AudioLM (Borsos et al., 2023) employs these tokens not only to produce realistic speech but also music. However, as acoustic tokens need to preserve a lot of information unrelated to semantics, their bitrates surpass the capacity of conventional language models.Consequently, handling these tokens requires specialized techniques (van den Oord et al., 2017; Borsos et al., 2023), making their practical utility challenging.

**Vector Quantization.** Learning vector quantization is important for efficient coding of information. VQ-VAE (van den Oord et al., 2017) introduces vector quantization into VAE (Kingma & Welling, 2014) to reconstruct images by learning discrete codebooks. They propose a straight-through gradient method to allow gradient back-propagation through a non-differentiable quantization operation so that optimization of the network is feasible. Furthermore, recent advancements include softmax quantization (Kankanahalli, 2018), exponential moving average (EMA) (Gârbacea et al., 2019), and Gumble-softmax (Yang et al., 2022) also work well for optimization of the VQ module. In addition to multiple VQ codebooks in van den Oord et al. (2017), Soundstream (Zeghidour et al., 2022) introduces a new RVQ method that is able to compress the raw audio with different bitrates.

## 3 METHOD

Despite the wide applications of semantic tokens in speech modeling (Borsos et al., 2023; Lee et al., 2022b), the discretization of the representations results in severe information loss. Consequently, downstream tasks, such as ASR or speech translation, suffer from a significant downgrade in performance. In AudioLM (Borsos et al., 2023), the WER is dramatically increased from 2.5% to 6.0% by using the discrete tokens of k-means from w2v-BERT XL (Chung et al., 2021). The unit-vocoder in mHuBERT (Lee et al., 2022b) also relatively increases the WER of the generated audio by about 70%. These results all demonstrate that the severe information loss actually prevents the discretization of speech from obtaining SOTA performance.

The information loss motivates us to preserve more information during the discretization of the representations. To this end, we propose a novel method, RepCodec, to perform more efficient compression on the representations so that the semantic tokens can preserve more information and achieve better performance in downstream tasks.

### 3.1 ARCHITECTURE OF REPCODEC

In order to achieve better compression of the representation, RepCodec uses a parametric network, which consists of 3 components (Figure 1): a codec encoder, a VQ module, and a codec decoder. The codec encoder takes as input the speech representations $\mathbf{X} = [\mathbf{x}_1, \mathbf{x}_2, \cdots, \mathbf{x}_T] \in \mathbb{R}^{H \times T}$ and produces latent representations $\mathbf{Z} = [\mathbf{z}_1, \mathbf{z}_2, \cdots, \mathbf{z}_T] \in \mathbb{R}^{H \times T}$. Here, $H$ is the dimension of the speech representation and $T$ is the length of the sequence. These latent representations $\mathbf{Z}$ are then passed through the VQ module to be quantized into a sequence of discrete tokens $\mathbf{s} = s_1 s_2 \cdots s_T$ with codebook $\mathbf{E} = [\mathbf{e}_1, \mathbf{e}_2, \cdots, \mathbf{e}_K]$, where $K$ is a predetermined number of clusters. The codec decoder utilizes these tokens $\mathbf{E}$ to reconstruct the original speech representations.

**Encoder and Decoder.** The architecture of the encoder and decoder follows Zeghidour et al. (2022); Wu et al. (2023), which achieves great success in compressing audio signals. As shown in Figure 2, the encoder consists of several 1D convolution layers with convolution applied to the time dimension of the input representation $\mathbf{X}$. The encoder block contains residual path to allow better optimization of the network (He et al., 2016). The decoder has a similar design, which is also composed of several 1D convolution layers and residual paths. In our paper, we do not downsample or upsample in both the encoder and the decoder, and keep frequency of the representation the same as the input.

**Vector Quantizer.** Vector Quantizer aims to compress the latent representations $\mathbf{Z}$ to a series of discrete tokens $\mathbf{s}$. It projects the latent $\mathbf{z}$ to its closest codebook $\mathbf{e}_k$ and outputs $\mathbf{e}_k$ to the decoder. We adopt two kinds of quantizer, regular VQ (van den Oord et al., 2017) and RVQ (Zeghidour et al., 2022). RVQ is a $M$-layer quantizer where each layer quantizes the residual of the previous layer, and it is effective for compressing the audio signals. When $M = 1$, RVQ is equivalent to VQ.

### 3.2 TRAINING OBJECTIVE

Our training objectives consist of a reconstruction loss on $\mathbf{X}$, which aims to preserve as much input information as possible for downstream tasks, and a quantization loss to effectively train the VQ.

**Reconstruction loss $l_r$.** We minimize the squared $\ell_2$ distance between the input representations $\mathbf{X}$ and the output representations $\hat{\mathbf{X}}$. Formally,

$$l_r = \frac{1}{HT} \|\mathbf{X} - \hat{\mathbf{X}}\|_F^2 \tag{1}$$

where $H$ is the hidden dimension of the representation and $\| \cdot \|_F$ denotes the Frobenius norm.

**Quantization loss $l_q$.** Following Défossez et al. (2022); Wu et al. (2023), we apply a quantization loss $l_q$ between the output of the encoder and the quantized value from VQ. Formally, given the latent representations $\mathbf{Z} = [\mathbf{z}_1, \mathbf{z}_2, \cdots, \mathbf{z}_t]$ and the codebooks $\mathbf{E} = [\mathbf{z}_1, \mathbf{z}_2, \cdots, \mathbf{z}_K]$, we minimize

$$l_q = \frac{1}{T} \sum_{t=1}^{T} \frac{1}{H} \sum_{k=1}^{K} \mathbb{I}_k(\mathbf{z}_t) \|\mathbf{z}_t - \mathbf{e}_k\|_2^2 \tag{2}$$

where $\mathbb{I}_k(\mathbf{z}_t) \in \{0, 1\}$ is binary indicator variables indicating which of the $K$ clusters the data point $\mathbf{z}_t$ is assigned to. $\mathbb{I}_k(\mathbf{z}_t) = 1$ if $\mathbf{z}_t$ is assigned to cluster $k$, and $\mathbb{I}_k(\mathbf{z}_t) = 0$ otherwise. When RVQ is

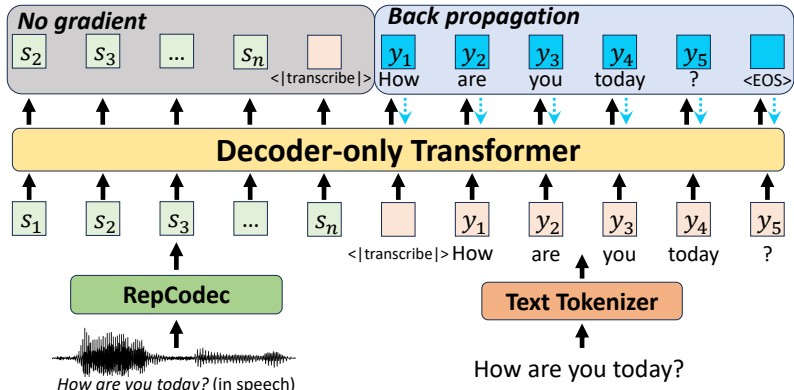

**Figure 3:** Illustration of decoder-only ASR using decoder-only Transformer architecture. Speech is tokenized by speech tokenizers and text is tokenized by SentencePiece (Kudo & Richardson, 2018). During training, we only compute gradients and apply back propagation on the text tokens (in blue).

used, the quantization loss in Equation (2) is generalized as:

$$l_q = \sum_{i=1}^{M} \frac{1}{T} \sum_{t=1}^{T} \frac{1}{H} \sum_{k=1}^{K} \mathbb{I}_k^i(\mathbf{z}_t^i)\|\mathbf{z}_t^i - \mathbf{e}_k^i\|_2^2 \tag{3}$$

where $i \in [1, M]$, denotes the $i^{th}$ quantizer of RVQ. And the superscript $i$ in $\mathbb{I}_k^i, \mathbf{z}_t^i, \mathbf{e}_k^i$ denotes the indicator, input representation and codebook for $i^{th}$ quantizer respectively. $l_q$ is only used for updating the encoder parameters. It makes the latent representation $\mathbf{Z}$ of the encoder suitable for the clustering of quantizer. The quantizer is updated by EMA described in Section 3.3.

Overall, RepCodec is trained by a weighted sum of the two losses,

$$l = \lambda_r \cdot l_r + \lambda_q \cdot l_q \tag{4}$$

## 3.3 Optimization of Vector Quantizer

Both k-means (Lloyd, 1982) and VQ (Gray, 1984) are algorithms to discretize a high-dimensional vector into a discrete label. These methods share the optimization of a common objective function: they both aim to find the best clusters measured by $\ell_2$ in Equation (2) (Bishop & Nasrabadi, 2006). However, they adopt different kinds of optimization algorithms.

K-means adopts an expectation-maximization (EM) algorithm (Bishop & Nasrabadi, 2006) to search for the best clusters. Despite its success in clustering raw input, its sharp changes hinder the back-propagation of gradient through the quantization module, which could potentially result in training process instability. On the contrary, the optimization algorithms adopted in VQ, including Straight-through Gradient Method, Exponential Moving Average (EMA) and Gumble-softmax, gradually change the quantization. This ensures a stable update of the encoder so that it can be trained end-to-end with other components of the model (van den Oord et al., 2017). In our paper, we follow the optimization in Zeghidour et al. (2022) and use the EMA algorithm for the optimization process. Formally speaking, let $\{\mathbf{z}_1, \ldots, \mathbf{z}_b\}$ be the minibatch input, where $b$ is the batch size, then the codebook entries $\mathbf{e}_k$ are updated by EMA with factor $0 \leq \gamma \leq 1$, where $\tilde{n}_k$ and $e_k$ represents the moving average of the number and codebook of the $k$-th cluster.

$$\tilde{n}_k = \gamma\tilde{n}_k + (1-\gamma)\sum_{j=1}^{b} \mathbb{I}_k(\mathbf{z}_j), \quad \tilde{\mathbf{e}}_k = \gamma\tilde{\mathbf{e}}_k + (1-\gamma)\sum_{j=1}^{b} \mathbb{I}_k(\mathbf{z}_j)\mathbf{z}_j, \quad \mathbf{e}_j = \frac{\mathbf{e}_i}{\tilde{n}_k}. \tag{5}$$

## 3.4 Downstream Tasks

To measure the performance of semantic tokens, we evaluate these tokens on downstream tasks of decoder-only ASR and unit-to-speech resynthesis. These two sets of experiments simulate the audio

input and audio output of a language model respectively. And we measure the amount of semantic information captured by the tokens by WER.

**Decoder-only ASR.** Following the emergence of LLM, and a series of works that inject audio information into a decoder-only transformer, we evaluate the quality of RepCodec using a decoder-only transformer. As shown in Figure 3, given a series of semantic audio tokens $\mathbf{s} = s_1 s_2 \cdots s_T$ and its corresponding transcript $\mathbf{y} = y_1 y_2 \cdots y_m$, we form a sequence

$$\mathbf{p} = \mathbf{s} \text{<|transcribe|>} \mathbf{y} = s_1 s_2 \cdots s_n \text{<|transcribe|>} y_1 y_2 \cdots y_m \qquad (6)$$

where $\text{<|transcribe|>}$ is a special token indicating the start the transcription. As ASR is a sequence-to-sequence task, we find a transformer $F$ that maximizes the conditional probability

$$F_* = \arg\max_F p(\mathbf{y}|\mathbf{s}) = \arg\max_F \prod_{i=1}^{m} p(y_i|y_{<i}, \mathbf{s}), \qquad (7)$$

instead of full language modeling of $p(\mathbf{s}, \mathbf{y})$. We also compare the full language modeling of $p(\mathbf{s}, \mathbf{y})$ in Appendix C.2, which is much worse than $p(\mathbf{y}|\mathbf{s})$. We implement the beam search as the decoding strategy with size 5. For convenience, we do not add additional language model or length penalty.

**Speech Resynthesis.** For each set of tokens, a unit-based HiFi-GAN vocoder is built to resynthesize speech. We follow Lee et al. (2022a) and Polyak et al. (2021) for the training and inference of the vocoders. The vocoders are trained with a combination of the generator-discriminator loss and the mean square error (MSE) of each unit segment in logarithmic domain. Following the common practice to evaluate semantic tokens (Borsos et al., 2023; Lee et al., 2022b), the quality of tokens are measured by ASR-WER of the resynthesized speech with the Whisper large-v2 model (Radford et al., 2023).

## 4 EXPERIMENTS

### 4.1 EXPERIMENT SETUPS

**Choices of Representation.** We select the widely-used self-supervised pretrained models, HuBERT (Hsu et al., 2021) and data2vec (Baevski et al., 2022), as the speech encoder of RepCodec. In accordance with the common practices of selecting the layer for representations, we choose the output from the layer at about 2/3 of the total layers as the input representations of RepCodec. In addition, we include the most-powerful open-sourced representations, Whisper (Radford et al., 2023), in our evaluation of RepCodec. As we find the representations from the top layer of Whisper encoder is most suitable for ASR in SUPERB (wen Yang et al., 2021), we use them as the input representations for RepCodec. Moreover, we extend our evaluation beyond single-layer representations, exploring whether the linear combinations of multiple layers are more suitable for the downstream tasks. We use SUPERB toolkit to find the best linear combination of representations for SUPERB ASR, and use it as the input representation for RepCodec.

**Baselines.** We compare RepCodec with several baselines, including **k-means** (Hsu et al., 2021), **VQ** and **EnCodec** (Défossez et al., 2022). **K-means** has been a predominant method in prior literature for semantic token extraction (Hsu et al., 2021; Lee et al., 2022b; Borsos et al., 2023). **VQ** directly takes the input speech representation $\mathbf{X}$ to the vector quantizer without either encoder or decoder. K-means and VQ are similar methods, sharing the same model and objective function, except that VQ uses the same optimization method as RepCodec. EnCodec is an open-source model similar to SoundStream (Zeghidour et al., 2022), which compresses the information directly from raw audio. We limit the bitrates of EnCodec for fair comparison among different methods.

In addition to the baselines, we also include ideal upper bound for decoder-only ASR and speech resynthesis respectively. For ASR, we replace the discrete tokens with the original representations for the input of the decoder, preserving the complete information for ASR. Concerning speech resynthesis, ASR-WER of the original audio is reported as a benchmark.

**Training Semantic Tokenizers.** The detailed architecture and hyperparameters of training RepCodec is available at Appendix B. We employ a fixed cluster count of $K = 1024$ for all semantic tokenizers. And we use the train-clean-100 subset of the LibriSpeech corpus (Panayotov et al., 2015)

**Table 1:** Main results on decoder-only ASR tasks. The WER scores are evaluated on the test-clean set of LibriSpeech. K-means, VQ, and RepCodec are trained on the train-clean-100 subset. Then we use these speech tokenizers to generate tokens for the entire 960h LibriSpeech. All the ASR models are Base transformer decoder-only models and are trained on 960h of representations or tokens.

| Representation | Multiple Layers (Linear Combination) | | | | | | Single Layer | | | | | |
|---|---|---|---|---|---|---|---|---|---|---|---|---|
| | HuBERT | | data2vec | | Whisper | | HuBERT | | data2vec | | Whisper | |
| | base | large | base | large | medium | large | base | large | base | large | medium | large |
| Method | - | - | - | - | - | - | 9th | 18th | 6th | 18th | 24th | 32nd |
| *Representation* | *3.62* | *2.91* | *3.06* | *2.18* | *4.54* | *6.16* | *4.02* | *2.81* | *3.77* | *2.18* | *3.94* | *3.96* |
| EnCodec (1RVQ 0.75kbps) | | | | | | 35.44 | | | | | | |
| EnCodec (2RVQ 1.5kbps) | | | | | | 16.53 | | | | | | |
| k-means (0.5kbps) | 10.83 | 6.14 | 6.57 | 7.23 | 100+ | 100+ | 6.36 | 5.00 | 5.97 | 4.55 | 9.52 | 9.97 |
| VQ (0.5kbps) | 10.20 | 5.17 | 6.14 | 8.53 | 100+ | 100+ | 6.27 | 5.19 | 6.20 | 4.68 | 24.35 | 44.43 |
| RepCodec (0.5kbps) | **9.93** | **4.11** | **4.87** | **5.39** | **12.89** | **13.12** | **5.73** | **4.02** | **5.15** | **2.87** | **5.04** | **5.01** |

**Table 2:** WER of ASR Modelling when scaling RepCodec, using RVQ, varying the number of clusters and applying to different languages.

**(a)** WER of scaled RepCodec.

| | HuBERT large 18th | data2vec large 18th |
|---|---|---|
| RepCodec (100h) | 4.03 | 2.87 |
| RepCodec (960h) | **3.72** | **2.65** |

**(b)** WER of RepCodec using RVQ.

| | HuBERT large 18th | data2vec large 18th |
|---|---|---|
| RepCodec (1 VQ) | 4.03 | 2.87 |
| RepCodec (2 RVQ) | **3.85** | **2.48** |

**(c)** WER of different number of clusters. We use HuBERT large 18th for the analysis.

| Clusters $K$ | 512 | 1024 | 2048 | 4096 |
|---|---|---|---|---|
| k-means | 5.50 | 5.00 | 4.71 | 4.78 |
| VQ | 5.39 | 5.19 | 4.54 | 4.71 |
| RepCodec | **4.14** | **4.02** | **3.89** | **4.03** |

**(d)** WER of speech in different languages . We use mHuBERT 11th for the analysis.

| Language | English | French | Spanish |
|---|---|---|---|
| k-means | 9.60 | 13.72 | 10.70 |
| VQ | 10.55 | 13.83 | 10.42 |
| RepCodec | **8.52** | **12.90** | **9.78** |

to train our all semantic tokenizers. It ensures a fair comparison between k-means, VQ and Rep-Codec (previous implementation of k-means cannot use large amount of audio data due to memory constraint). For multilingual experiments, we further incorporate 100h subsets from MLS (Pratap et al., 2020) for French and Spanish.

## 4.2 DECODER-ONLY ASR

We convert the full 960h of LibriSpeech (Panayotov et al., 2015) speech into speech tokens and use them to train decoder-only ASR models. The decoder is a Base transformer with 12 layers, embedding dimension 768 and FFN dimension 3072. After training, the ASR models are then evaluated at the test-clean and dev-clean subsets of LibriSpeech. Detailed experimental setup are deferred to Appendix B.

As shown in Table 1, RepCodec achieves much lower WER than both k-means and VQ across all representations. For single-layer representations, RepCodec is particularly effective for large speech encoders such as data2vec large and Whisper. For data2vec large, RepCodec improves WER by about 2% in absolute value, and achieves very close performance to the original representation. In case of both Whisper medium and Whisper large models, RepCodec improves the WER by more than 4% in absolute terms, relatively decreasing WER by nearly 50%.

For linear combination of representations, we observe that they are not as suitable as the single layer representations for clustering. Nevertheless, RepCodec still achieves large improvement in WER. Particularly, RepCodec is able to produce meaningful WER for Whisper representations, while both VQ and k-means cannot successfully cluster them. Although EnCodec (Défossez et al., 2022) preserves information beyond linguistic content of the speech, its performance is inferior to RepCodec in terms of semantic information. While EnCodec uses higher bitrates than RepCodec (1.5kps v.s 0.5kbps), RepCodec still achieves lower WER. It shows that RepCodec is more suitable for downstream tasks which rely on semantic information of speech.

**Scaling RepCodec.** In Table 2a, we use all 960h data of LibriSpeech to train a larger RepCodec model (the architecture is deferred to Appendix A). The downstream decoder-only ASR shows that the model achieves even lower WER, validating the scaling ability of our method.

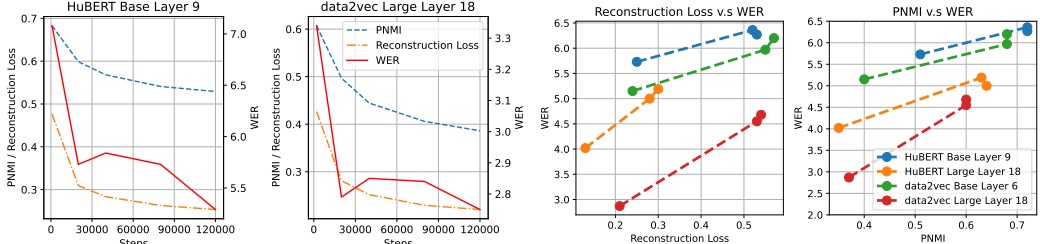

**Figure 4:** Left: Changes of PNMI, reconstruction loss $l_r$ and WER of decoder-only ASR on test-clean of LibriSpeech as the training step of RepCodec increases. Right: Relationship between of PNMI, reconstruction loss $l_r$, and WER of decoder-only ASR on k-means, VQ and RepCodec.

**Residual Vector Quantization.** In Table 2b, we present the performance of RepCodec with a 2-layer RVQ. It has higher bitrates and preserve more information of the speech representation. Therefore, RepCodec has further improvements in the downstream decoder-only ASR task.

**Multilingual.** We conduct experiments with representations from mHuBERT (Lee et al., 2022b) and train a unified speech tokenizer for three languages (English, French and Spanish). Then we train a unified decoder for ASR of all three languages. The WER is evaluated on the test-clean of LibriSpeech and the test sets of MLS. As shown in Table 2d, RepCodec outperforms k-means and VQ across all three languages, demonstrating that RepCodec can be applied to multiple languages.

## 4.3 SPEECH RESYNTHESIS

We conduct the unit-to-speech resynthesis on two datasets: LJSpeech (Ito & Johnson, 2017) and VCTK (Veaux et al., 2017) . LJSpeech is a single speaker English TTS corpus comprising 13,100 speech utterances, equivalent to approximately 24 hours of audio. VCTK is a multi-speaker English TTS corpus uttered by 109 speakers. It comprises around 43,800 speech utterances, equivalent to approximately 44 hours of audio. We follow the data partition in Polyak et al. (2021), and split the data into training sets, validation sets and test sets. All the audios are downsampled to 16kHz and trained with a fixed 50k training steps. For the experiments of LJSpeech, we use the architecture and toolkit provided in Lee et al. (2022a) for the training and inference of the vocoders. For VCTK, we follow the model architecture and toolkit provided in Polyak et al. (2021). In this setup, the synthesized speech's speaker characteristics are conditioned on speaker embeddings. In Table 3b, "Single" indicates that the speaker embedding utilized for resynthesizing the speech corresponds to the ground truth speaker. On the other hand, "VC" refers to voice conversion, where speaker embeddings from two male and two female speakers were randomly selected from the seen speakers.

In Table 3, we report the ASR-WER of EnCodec, RepCodec and k-means. We only compare RepCodec with k-means, which is similar to VQ and achieves lower WER in ASR. We only evaluate semantic tokens of single-layer representations, which are shown more suitable for downstream tasks in Table 1. When using Whisper to transcribe the speech, we turn off the temperature scheduling and use temperature 0 to remove the randomness of the evaluation. The WER difference between original audio and resynthesized speech shows the quality of the semantic tokens.

RepCodec reduces WER by more than 2% in absolute value for all these representations in both LJSpeech and VCTK, which is much more significant than the improvement in decoder-only ASR. For Whisper representations, RepCodec improves WERs by more than 30% in absolute terms. This performance is in stark contrast to the deteriorations of ASR-WER in Borsos et al. (2023) and Lee et al. (2022b). In those instances, the WER experienced a relative increase of approximately 70%. However, for RepCodec, this relative downgrade is significantly reduced to about 35%.

## 4.4 ANALYSIS

**Phone-Normalized Mutual Information (PNMI) versus Reconstruction Loss.** Hsu et al. (2021) and Borsos et al. (2023) propose several methods to measure the quality of semantic tokens, including PNMI and ABX error. These quantities measure the similarity between phonemes and semantic tokens. However, the observations presented in our work diverge from the hypothesis that token sets with higher similarities lead to better performance.

**Table 3:** ASR-WER of the resynthesized speech of test set of LJSpeech and VCTK. The ASR-WER is computed with Whisper large-v2.

**(a)** LJSpeech

| Method / Representation | HuBERT large 18th | data2vec large 18th | Whisper medium 24th | Whisper large 32nd |
|---|---|---|---|---|
| *Original Audio* | | | *3.44* | |
| EnCodec (1RVQ 0.75kbps) | | | 14.70 | |
| EnCodec (2RVQ 1.5kbps) | | | 9.74 | |
| k-means (0.5kbps) | 7.61 | 9.90 | 36.02 | 100+ |
| RepCodec (0.5kbps) | **4.71** | **5.25** | **5.62** | **6.18** |

**(b)** VCTK.

| Method / Representation | HuBERT large 18th Single | HuBERT large 18th VC | data2vec large 18th Single | data2vec large 18th VC | Whisper large 32nd Single | Whisper large 32nd VC |
|---|---|---|---|---|---|---|
| *Original Audio* | | | *3.28* | | | |
| EnCodec (1RVQ 0.75kbps) | | | 52.67 | | | |
| EnCodec (2RVQ 1.5kbps) | | | 10.13 | | | |
| k-means (0.5kbps) | 6.32 | 6.61 | 10.91 | 10.21 | 35.24 | 38.8 |
| RepCodec (0.5kbps) | **4.58** | **4.41** | **4.88** | **4.61** | **6.43** | **7.19** |

In the left plots of Figure 4, we show the changes of PNMI, reconstruction loss and WER as the training step increases. The reconstruction loss is normalized by its $\ell_2$ norm. The reconstruction loss decreases as the training of RepCodec proceeds, and so does the WER of decoder-only ASR. However, PNMI also decreases for longer training steps. The right of Figure 4 shows a similar observation, where we plot the relationship of reconstruction loss v.s WER and PNMI v.s WER for k-means, VQ and RepCodec for different speech representations. Methods with higher PNMI do not result in lower WER in downstream tasks. In contrast, downstream tasks performance is positively correlated to the reconstruction loss of the clustering. These outcome underscores that higher PNMI does not necessarily correspond to reduced WER values. Instead, when we increasingly retain information from the speech representation (represented by decreasing $l_r$), the semantic tokens have higher quality for downstream tasks, although these tokens get dissimilar to the phonemes.

It is worth noting that our findings do not contradict to the assertion made by Hsu et al. (2021), which suggests token sets with higher PNMI lead to better performance. In Hsu et al. (2021), the discretized tokens serve as training targets for the speech encoders, while our tokens serve as representations of the speech itself for downstream tasks. The difference in token usages leads to the diversion on the perspective of PNMI.

**Number of Clusters.** In Table 2c, we study how the performance of RepCodec for varying number of clusters $K$. With different $K$, RepCodec all outperforms k-means and VQ. Moreover, RepCodec is more robust against the changes of $K$ than other two baselines. Even RepCodec with $K = 512$ has lower WER than k-means with $K = 4096$.

## 5 CONCLUSION

Interacting with LLMs through speech leads to an increased demand for speech tokenization. To this end, we propose RepCodec, a novel speech representation codec to convert continuous speech waveforms into discretized tokens. In contrast to previous methods, RepCodec employs a parametric network to preserve more semantic information of the speech representations. The extensive experiments demonstrate that semantic tokens extracted RepCodec outperform the prevalent k-means algorithm in downstream tasks of both speech understanding and generation. Moreover, the experiments also demonstrate that RepCodec is a universal algorithm that can be applied to any speech encoders and to multiple languages.

While our method obviously outperforms the baselines, challenges still remain. There is still a performance gap between using representations and discretized tokens. Future works may minimize

the gap with more sophisticated codec architectures (*e.g* Transformers) and objective functions (*e.g.* adversarial loss). Furthermore, the framework RepCodec may also be extended to other tasks like speech translation and video understanding.

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

# A DETAILED ARCHITECTURE OF REPCODEC

In this section, we will introduce the architecture of RepCodec. We first provide the components of RepCodec and then show the specific architecture used for out method in Table 4.

**1D Convolution Layer `Conv1d(C, c, k, s)`.** We denote `Conv1d(C, c, k, s)` as the non-causal 1D convolution layer with input channel $C$, output channel $c$, kernel size $k$ and stride $s$ without dilation.

**Residual Unit `Res(C, k, s)`.** We denote `Res(C, k, s)` as the residual unit used for both the encoder block and decoder block in Figure 2. Each `Res(C, k, s)` consists of two `Conv1d(C, C, k, s)` layers, with a residual path added across them.

**Encoder Block `Enc(C, k, s)`.** With the input having a channel of $C$, the encoder block `Enc(C, k, s)` consists of two Residual Units followed by a 1D convolution layer:

`Res(C, k, s) -> Res(C, k, s) -> Conv1d(C, C, k, s)`

**Decoder Block `Dec(C, k, s)`.** With the input having a channel of $C$, the decoder block `Dec(C, k, s)` consists of a 1D convolution layer followed by two Residual Units:

`Conv1d(C, C, k, s) -> Res(C, k, s) -> Res(C, k, s)`

**Residual Vector Quantizer `RVQ(M, K, C)`.** `RVQ(M, K, c)` denotes a $M$-layer Residual Vector Quantizer (RVQ) with number of clusters $K$ and codebook dimension $C$. When $M = 1$, RVQ is equivalent to VQ.

**Table 4:** Architecture of RepCodec. $H$ is the dimension of the corresponding speech representation.

|                    | Regular RepCodec | RepCodec in Table 2b | RepCodec in Table 2a |
|--------------------|------------------|----------------------|----------------------|
|                    | `Conv1d(H, H, 3, 1)` | `Conv1d(H, H, 3, 1)` | `Conv1d(H, H, 3, 1)` |
| Encoder            | `Enc(H, 3, 1) ×2` | `Enc(H, 3, 1) ×2` | `Enc(H, 3, 1) ×8` |
|                    | `Conv1d(H, H, 3, 1)` | `Conv1d(H, H, 3, 1)` | `Conv1d(H, H, 3, 1)` |
| Vector Quantizer   | `RVQ(1, 1024, H)` | `RVQ(2, 1024, H)` | `RVQ(1, 1024, H)` |
|                    | `Conv1d(H, H, 3, 1)` | `Conv1d(H, H, 3, 1)` | `Conv1d(H, H, 3, 1)` |
| Decoder            | `Dec(H, 3, 1) ×2` | `Dec(H, 3, 1) ×2` | `Dec(H, 3, 1) ×2` |
|                    | `Conv1d(H, H, 3, 1)` | `Conv1d(H, H, 3, 1)` | `Conv1d(H, H, 3, 1)` |

# B DETAILS OF THE EXPERIMENTS

## B.1 TRAINING SPEECH TOKENIZERS.

**K-means.** We use the script from HuBERT[1] to train the k-means model and perform k-means clustering, with all hyperparemeters unchanged.

**RepCodec.** We train RepCodec for 200,000 steps. The batch size is 32 speech representations, each of which has 96 frames. We use Adam (Kingma & Ba, 2014) to optimize the model with a fixed learning rate $1 \times 10^{-4}$ and $\beta_1 = 0.5, \beta_2 = 0.9$. We set $\lambda_r = 45, \lambda_q = 1$ and weight decay as 0 for all the experiments.

**VQ.** We remove the encoder and decoder from RepCodec and train the model for 50,000 steps. Other hyperparameters are the same as RepCodec.

## B.2 DECODER-ONLY ASR.

We use the Base Transformer decoder from fairseq (Ott et al., 2019) for decoder-only ASR. We fix the training steps to 100,000 for all the experiments. The decoder is optimized by Adam with $\beta_1 = 0.9, \beta_2 = 0.999$. The learning rate is warmed up by 5,000 steps to $1 \times 10^{-3}$ and then follow inverse square root decay to 0. We use cross entropy loss with smooth factor 0.1 to optimize the model, and we select the best checkpoint on the dev-clean set for further evaluation on the test sets.

---

[1] `https://github.com/facebookresearch/fairseq/tree/main/examples/hubert/simple_kmeans`

**Table 5:** The WER on the dev-clean subset of LibriSpeech.

| Representation / Method | Multiple Layers (Linear Combination) | | | | | | Single Layer | | | | | |
| | HuBERT | | data2vec | | Whisper | | HuBERT | | data2vec | | Whisper | |
| | base | large | base | large | medium | large | base | large | base | large | medium | large |
| | - | - | - | - | - | - | 9th | 18th | 6th | 18th | 24th | 32nd |
| *Representation* | 3.27 | 2.76 | 2.79 | 2.05 | 4.16 | 5.39 | 3.90 | 2.65 | 3.37 | 2.13 | 3.74 | 3.58 |
| EnCodec (1RVQ 0.75kbps) | | | | | 36.39 | | | | | | | |
| EnCodec (2RVQ 1.5kbps) | | | | | 16.78 | | | | | | | |
| k-means (0.5kbps) | 10.91 | 5.32 | 6.39 | 7.36 | 100+ | 100+ | 6.04 | 4.99 | 6.26 | 4.59 | 9.68 | 10.15 |
| VQ (0.5kbps) | 10.59 | 4.98 | 5.92 | 8.46 | 100+ | 100+ | 6.02 | 5.18 | 6.80 | 4.84 | 34.57 | 44.29 |
| RepCodec (0.5kbps) | **9.38** | **3.96** | **4.71** | **5.01** | **12.03** | **12.36** | **5.40** | **3.76** | **5.04** | **2.12** | **4.72** | **4.75** |

**Table 6:** WER of dev sets of multilingual experiments.

| Language | English | French | Spanish |
| --- | --- | --- | --- |
| k-means | 9.70 | 15.89 | 10.21 |
| VQ | 10.76 | 15.65 | 10.35 |
| RepCodec | **8.78** | **14.97** | **9.42** |

We use SentencePiece as the text tokenizer, and we train a new SentencePiece model on each dataset with all its transcripts. The vocabulary size is 5,000 for English dataset. For multilingual dataset in Table 2d, we jointly train the SentencePiece model of all three languages with a vocabulary size of 10,000.

## C  ADDITIONAL EXPERIMENTS

### C.1  ADDITION RESULTS OF DECODER-ONLY ASR.

Table 5 and Table 6 show the WER of ASR on the dev sets. RepCodec still outperforms baselines by a large margin. It is worth noting the RepCodec achieves even lower WER on the layer 18 representation of data2vec large.

**Table 7:** The WER on the dev-clean subset of LibriSpeech.

| | HuBERT large 18th | | data2vec large 18th | |
| --- | --- | --- | --- | --- |
| | test-clean | dev-clean | test-clean | dev-clean |
| k-means $p(\mathbf{y}|\mathbf{s})$ | 5.00 | 4.99 | 4.55 | 4.59 |
| k-means $p(\mathbf{s}, \mathbf{y})$ | 7.17 | 7.10 | 9.45 | 8.47 |
| VQ $p(\mathbf{y}|\mathbf{s})$ | 5.19 | 5.18 | 4.68 | 4.84 |
| VQ $p(\mathbf{s}, \mathbf{y})$ | 8.85 | 8.65 | 8.32 | 8.05 |
| RepCodec $p(\mathbf{y}|\mathbf{s})$ | 4.02 | 3.76 | 2.87 | 2.12 |
| RepCodec $p(\mathbf{s}, \mathbf{y})$ | 6.70 | 6.47 | 6.77 | 6.30 |

### C.2  FULL LANGUAGE MODELLING V.S CONDITIONAL LANGUAGE MODELLING

Table 7 compares the WER on the test-clean and dev-clean subset of LibriSpeech when optimizing $p(\mathbf{s}, \mathbf{y})$ or $p(\mathbf{y}|\mathbf{s})$ for decoder-only ASR. For all three methods, full language modelling $p(\mathbf{s}, \mathbf{y})$ results in much higher WER than the conditional language modelling $p(\mathbf{y}|\mathbf{s})$. Therefore, we choose to optimize $p(\mathbf{y}|\mathbf{s})$ for decoder-only ASR task.

### C.3  ABLATION STUDY ON $\lambda_q$ AND $\lambda_r$

As shown in Table 8, we change the weights of reconstruction loss $\lambda_r$ to train the encoder of Rep-Codec and report its WER on the downstream task of ASR modeling with representation from the

**Table 8:** WER, PNMI and Reconstruction Loss of decoder only ASR modeling from 18th layer of large data2vec model.

| $\lambda_q$ | $\lambda_r$ | PNMI | Reconstruction Loss | WER |
|---|---|---|---|---|
| 1.0 | 30.0 | 0.3689 | 0.2091 | 2.91 |
| 1.0 | 45.0 | 0.3714 | 0.2100 | 2.87 |
| 1.0 | 60.0 | 0.3697 | 0.2090 | 2.83 |

18th layer of data2vec large model. The results show that RepCodec is robust to the changes of the weights of $(\lambda_q, \lambda_r)$.

