# OpenReview forum: "RepCodec: A Speech Representation Codec for Speech Tokenization"
_ICLR.cc/2024/Conference — Submitted to ICLR 2024_

### Official Review · Reviewer_gJKh · 2023-10-24

**Soundness:** 3 good
**Presentation:** 3 good
**Contribution:** 2 fair
**Rating:** 6
**Confidence:** 3

**Summary:**

The authors proposed a new speech tokenization for semantic modeling in this paper. Previous methods usually use k-means to discrete semantic representation, leading to information loss. Inspired by the audio codecs which reconstruct the raw audio, RepCodec encodes speech semantic representation from Hubert or data2vec to a set of vector quantization codebooks and reconstructs them by a decoder. The experiments demonstrate the superior performance of RepCodec in speech understanding and generation. Many detailed experiments also evaluate the performance in different configurations.

**Strengths:**

The motivation and description of the proposed method are very clear and easy to understand. The experiment is sufficient and demonstrates the effectiveness of RepCodec.

**Weaknesses:**

1. In the introduction, the term "semantic quality" is mentioned. For clarity and the benefit of readers, could you provide a definition on what "semantic quality" encompasses?
2. In previous works such as VALLE, AuidoLM, and PolySpeech mentioned in this paper, they all use the k-means clustering method to obtain semantic tokens. For the speech generation task (TTS, VC), the semantic information is important in discrete tokens, but the other
encoded information such as speaker timbre is harmful to the task. Given the importance of speaker timbre information for downstream speech generation tasks, it would be of great value if a verification for speaker information of the encoded features is incorporated.

**Questions:**

1. Section II.Speech Tokenization. "However, the discretization step of k-means discards plenty of information of the speech“. Can you give some examples so that readers can better understand the limitations of k-means?
2. The first paragraph in Sectiion III. "In AudioLM (Borsos et al., 2023), the WER is dramatically increased from 2.5% to 6.0% by using the discrete tokens of k-means from w2v-BERT XL". What does this number 2.5% refer to?
3. In Equation 5, some symbols are not defined.
4. One question not related to the proposed method.  Why does the performance gap change among different speech encoders after VQ and K-means quantization compared with the original speech representations, especially whisper.  In other words, what kind of representations are suitable for clustering?
5. Can you provide some samples of the speech resynthesis?

---

> ### Author Response · Authors · 2023-11-23
> **Response to Reviewer gJKh**
>
> Thanks for the valuable suggestions from the reviewer. We respond to your questions as follows.
> - **“Meaning of semantic quality”**
>
>     We believe we do not directly mention  "semantic quality". Instead, we mention the “quality of semantic tokens”, which refers to suitability of the semantic tokens for downstream tasks like ASR or TTS. When integrating LLMs with semantic tokens, we need to discretize speech into tokens to perform different downstream tasks. Some discrete tokens may not be suitable for downstream tasks and leads to inferior downstream task performance.
> - **“Speaker information of RepCodec”**
>
>     We agree with the reviewer that the semantic units should carry as much semantic information as possible and as less speaker information as possible. Otherwise they may affect the speaker effect brought by other components, e.g. speaker embedding, acoustic tokens. We verify the speaker information carried by our units and the baseline k-means units through the voice conversion task:
>
>     | Methods                   | Speaker Similarity ↑| MOS↑ |
>     |:---|:---:|:---:|
>     |k-means(baseline)     |0.771                    |2.82±0.36|
>     |RepCodec                 |0.764                    |3.48±0.74|
>
>     In our voice conversion experiment, the speaker characteristics of the generated speech is controlled by the speaker embedding. However, the speaker information carried by the semantic units (if any) may also affect the result. In the above table, we measure the similarity of the voice converted speech and the original speech from the same speaker. A higher similarity score implies less speaker information carried by the units. It can be seen that the similarity scores are very close but RepCodec units have a much higher MOS score due to the improved semantic part. This means that RepCodec does not carry extra speaker information which is harmful to the task.
>
>
> - **“Limitation of k-means”**
>
>     The limitation of k-means is reflected in the degraded performance of the ASR modeling. Even if we train the seq-to-seq model with large amounts of data, using k-means tokens as input always results in much worse WER. The explanation to this empirical observation is that k-means discards some information of speech and makes the speech recognition of some similar words impossible so that the WER is inferior to using raw speech as input.
> - **“Referred number of 2.5%”**
>
>     We refer to Table II of the AudioLM paper, where WER of the original speech is 2.5%. However, the reconstructed speech using k-means has a much worse WER of 6.0%.
> - **Equation (5)”**
>
>     Thanks for your advice. We will add explanations to symbols $\tilde{n}_k$ and $e_k$ in our revision. They represent the moving average of EMA. $\tilde{n}_k$ represents the moving average of the number of clusters $k$ and $e_k$ represents the moving average of the code book of cluster $k$.
> - **“What kind of representations are suitable for clustering?”**
>
>     From our experiments, we believe that encoders trained with SSL objective function may be more suitable for clustering. Whisper is trained by end-to-end encoder-decoder architecture so that its representation from the encoder may require complicated transformation to be decoded as text. Therefore, its representation may be less suitable for clustering.
> - **“Examples of speech resynthesis”**
>
>     We have uploaded some resynthesized speech to supplementary material.

---

> > ### Comment · Reviewer_gJKh · 2023-11-23
> >
> > Thanks for your reply to my comments. I have no more questions. It is nice to see the optimization and analysis of clustering methods for semantic representation.

---

### Official Review · Reviewer_6oit · 2023-10-30

**Soundness:** 3 good
**Presentation:** 3 good
**Contribution:** 2 fair
**Rating:** 5
**Confidence:** 3

**Summary:**

In this paper, a speech representation code called RepCodec is introduced for semantic speech tokenization. It is seamlessly integrated into an end-to-end framework.

**Strengths:**

1. RepCodec demonstrates promising results in both ASR and unit-to-speech resynthesis compared to the clustering method.
2. The discovery that PMNI can deviate from performance is intriguing.

**Weaknesses:**

1. Overall, this paper lacks novelty, as compared to SouldStream, it simply replaces the input from raw waveform with SSL representations.
2. Some parts of the details in this paper are confusing:
    * The difference in bar height in the encoder and decoder parts in Figure 1 is confusing because neither sampling nor dimension reduction is applied.
    * Equation (5) lacks sufficient explanation. I am unsure of its correctness as neither ${\overset{\sim}{n_k}}$ nor $\mathbf{e}_{i}$ is adequately defined or explained.
    * Shouldn't equation (7) be
$ F^* = \arg\max_F p(\mathbf{y}|\mathbf{s}) = \arg\max_F \prod_{i=1}^{m} p(y_i|y_{<i}, \mathbf{s}) $?

3. It would be better to include a more in-depth analysis of the weights ($\lambda_{r}$, $\lambda_{q}$) of reconstruction loss and quantization loss.

**Questions:**

3. Is WER a common metric for speech resynthesis?

---

> ### Author Response · Authors · 2023-11-23
> **Response to Reviewer 6oit**
>
> Thanks for the feedback from the reviewer. We have the following response to your comments.
>
> - **“Novelty of the paper”**
>
>     We believe our paper proposes completely new methods to generate semantic units and provides new understanding of the quality of the semantic units. Before RepCodec, semantic units can only be produced by traditional k-means methods following HuBERT [1], which was proposed about two years ago. Our paper, despite technical similarity to SoundStream, provides a novel method that largely outperforms long existing k-means clustering.
>
>     In addition, we also provide a new understanding about the semantic units. In previous papers such as HuBERT [1] and AudioLM [2], similarity to phonemes is used to measure the quality of the semantic units. Our paper, however, shows that PNMI is not the only metric for evaluating the semantic tokens. Instead, the reconstruction loss provides more meaningful indication for the performance of downstream tasks. It brings new insight for future study of semantic tokens.
> - **“Ablation Study of $(\lambda_q, \lambda_r)$”**
>
>     As shown in the following Table, we change the weights of reconstruction loss $\lambda_r$ to train the encoder of RepCodec and report its WER on the downstream task of ASR modeling with representation from the 18th layer of data2vec large model. The results show that RepCodec is robust to the changes of the weights of $(\lambda_q, \lambda_r)$.
>     | $\lambda_q$ | $\lambda_r$ | PNMI |Reconstruction Loss| WER of ASR Modeling |
>     |:---|:---:|:---:|:---:|:---:|
>     |1.0|30.0| 0.3689 | 0.2091 | 2.91 |
>     |1.0|45.0| 0.3714 | 0.2100 | 2.87 |
>     |1.0|60.0| 0.3697 | 0.2090 | 2.83 |
>
> - **“Confusing details of the paper: Figure 1”**
>
>     Our method is compatible with different architectures of the encoder and decoder in Figure 1. To allow variants of the network architectures, we show in Figure 1 that dimension reduction can be applied. We will add more detailed illustrations in Figure 1.
> - **“Confusing details of the paper: Equation (5)”**
>
>     Thanks for your advice. We will add explanations to symbols $\tilde{n}_k$ and $e_k$ in our revision. They represent the moving average of EMA. $\tilde{n}_k$ represents the moving average of the number of clusters $k$ and $e_k$ represents the moving average of the code book of cluster $k$.
> - **“Confusing details of the paper: Equation (7)”**
>
>     Thanks for pointing out our typo. We will fix it in our revision.
> - **“Using WER to evaluate speech resynthesis”**
>
>     We follow previous papers and use WER to evaluate the quality of semantic units such as mHubert [3] and AudioLM [2]. As the role of semantic units is to provide the semantic information of the speech, the best way to evaluate the semantic information is to measure the content, i.e. ASR-WER, of the output speech.
>
> [1] Hsu, Wei-Ning, et al. "Hubert: Self-supervised speech representation learning by masked prediction of hidden units." IEEE/ACM Transactions on Audio, Speech, and Language Processing 29 (2021): 3451-3460.
>
> [2] Borsos, Zalán, et al. "Audiolm: a language modeling approach to audio generation." IEEE/ACM Transactions on Audio, Speech, and Language Processing (2023).
>
> [3] Lee, Ann, et al. "Textless Speech-to-Speech Translation on Real Data." In Proceedings of the 2022 Conference of the North American Chapter of the Association for Computational Linguistics: Human Language Technologies (2022).

---

> > ### Comment · Reviewer_6oit · 2023-11-23
> > **Increasing score from 3 to 5**
> >
> > I appreciate the authors' response. After considering their feedback and taking into account comments from other reviewers, I would like to revise my rating from 3 to 5.

---

### Official Review · Reviewer_wBo4 · 2023-11-01

**Soundness:** 2 fair
**Presentation:** 3 good
**Contribution:** 2 fair
**Rating:** 5
**Confidence:** 3

**Summary:**

This paper introduces RepCodec, a speech representation codec designed for semantic speech tokenization. It applies VQVAE to the representations from pretrained speech encoders to learn audio semantic tokens. The authors demonstrate the superiority of their proposed method over other discrete speech representation techniques, as evidenced by improved WER scores on ASR and speech resynthesis tasks.

**Strengths:**

* The authors demonstrate the superiority of their proposed method over other discrete speech representation techniques in terms of the WER scores on both ASR and speech resynthesis tasks.
* The authors analyze the issue with the quality measure of semantic tokens based on their similarity to ground truth phonemes, while illustrating that the reconstruction loss of their proposed method exhibits a higher correlation.

**Weaknesses:**

* Insufficient evaluation metrics. The research predominantly relies on WER as the principal evaluation metric for the performance of semantic speech tokens. To make a compelling case for the proposed method's superiority, it's essential to include other the evaluation metrics such as speaker similarity, F0 error, or mean-opinion score in the speech resynthesis experiments.
* Limited exploration of core downstream tasks. While semantic tokens are integral to token-based language modeling of speech, the paper's experiments are primarily focused on ASR and speech resynthesis. It lacks empirical investigations into other vital application tasks such as language modeling of audio, text-to-speech, speech-to-speech translation, or conditional modeling of acoustic tokens given the semantic tokens.

**Questions:**

I have concerns regarding the lack of evaluation results, as mentioned in the above weaknesses.

---

> ### Author Response · Authors · 2023-11-23
> **Response to Reviewer wBo4**
>
> Thanks for the constructive comments from the reviewer. We address your concern as follows.
>
> - **“Insufficient evaluation metrics.”**
>
>     First, our evaluation metrics follow previous papers on evaluating the quality of semantic units such as mHubert [1] and AudioLM [2]. They all use word error rate (WER) to measure the quality of semantic units. As the role of semantic units is to provide the semantic information of the speech, the best way to evaluate the semantic information is to measure the content, i.e. ASR-WER, of the output speech.  Other metrics such as speaker similarity or mean-opinion score (MOS) measure the acoustic quality of the speech, so they do not reflect the quality of the semantic units.
>
>     While we do not think speaker similarity and MOS can measure the quality of RepCodec units, we still perform the experiments to calculate the speaker similarity, F0 error, and mean-opinion score on speech resynthesis task using LJSpeech. As shown in the following Table, RepCodec largely outperforms k-means on all these metrics, proving the superiority of the acoustic quality of RepCodec units in the speech generation tasks.
>
>     | Methods|F0 VDE↓| F0 FFE ↓| Speaker Similarity ↑| MOS↑ |
>     |:---|:---:|:---:|:---:|:---:|
>     |k-means|0.199|0.248|0.771|2.82±0.36|
>     |RepCodec|0.174|0.185|0.764|3.48±0.74|
>     |Original Speech| - | - | - |4.32±0.92|
>
>
> - **“Insufficient downstream tasks.”**
>
>     First, we would like to point out that our experiments do focus on practical downstream tasks. In addition to ASR and speech resynthesis, we also perform voice conversion in Table 3(b). ASR is a critical task for speech understanding, and RepCodec can be corporated into frameworks like PolyVoice for speech-speech translation.  Voice conversion is also a practical task that has wide applications.
>
>     Completely performing the experiments like language modeling of audio, text-to-speech, speech-to-speech translation needs a lot of computation resources and a large amount of time to train the model, which makes it infeasible to complete the experiments in such a short time of the rebuttal. However, our experiments have verified both the input stage and output stage in the framework of PolyVoice for speech-to-speech translation, we believe it is sufficient to show that RepCodec is superior to k-means in these downstream tasks.
>
> [1] Lee, Ann, et al. "Textless Speech-to-Speech Translation on Real Data." In Proceedings of the 2022 Conference of the North American Chapter of the Association for Computational Linguistics: Human Language Technologies (2022).
>
> [2] Borsos, Zalán, et al. "Audiolm: a language modeling approach to audio generation." IEEE/ACM Transactions on Audio, Speech, and Language Processing (2023).

---

> ### Comment · Reviewer_wBo4 · 2023-11-23
> **Official Comment by Reviewer wBo4**
>
> I appreciate the authors for addressing my comments. However, since the focus of this study is not on more powerful self-supervised representation learning, but rather on producing effective discrete units from representations of self-supervised pretrained models, I believe it should have further validated its effectiveness in language modeling on speech data. Considering both the strengths and limitations of this research, I will maintain my current rating.

---

### Official Review · Reviewer_XznZ · 2023-11-01

**Soundness:** 3 good
**Presentation:** 4 excellent
**Contribution:** 2 fair
**Rating:** 6
**Confidence:** 4

**Summary:**

A novel RepCodec, a speech representation codec for semantic speech tokenization, has been introduced. RepCodec utilizes a vector quantization codebook to reconstruct speech representations from speech encoders like HuBERT or data2vec. RepCodec significantly outperforms the widely used k-means clustering approach in both speech understanding and generation tasks.

**Strengths:**

The paper is great in its clarity and well-structured organization. Its proposed approach is lauded for its simplicity and effectiveness. The comprehensive nature of the experiments conducted further strengthens the paper's credibility. Based on these positive aspects, it is recommended for publication at the conference.

**Weaknesses:**

The simplicity and effectiveness of the proposed approach are commendable. While there are no significant weaknesses to highlight, it would be intriguing to see the application of RepCodec in the context of zero-shot Text-to-Speech (TTS) systems, such as Vall-E. Exploring its potential in this domain could provide valuable insights and possibly further advancements in speech-processing technology.

The idea of SpeechTokenizer (SpeechTokenizer: Unified Speech Tokenizer for Speech Large Language Models) has some similarities, could you please elaborate more regarding the difference? If possible, adding some baseline numbers using https://github.com/ZhangXInFD/SpeechTokenizer would add value to this paper.

**Questions:**

See comment in the Weaknesses.

---

> ### Author Response · Authors · 2023-11-23
> **Response to Reviewer XznZ**
>
> We sincerely appreciate your positive review and the valuable feedback you have provided. We hope our response completely addresses any concerns you may have.
>
> - **“Zero-Shot TTS such as VALL-E”**
>
>     We agree with the reviewer that applying semantic units of RepCodec to zero-shot TTS is an interesting experiment. However, VALL-E does not use semantic units as the inputs for semantic information. Instead, they use phoneme sequence to control the semantic content of the output speech. Our experiments include both speech resynthesis and voice conversion tasks, which demonstrate superior performance to the traditional k-means clustering. We believe current experiments are enough to demonstrate the superiority of RepCodec, and we leave the experiments of zero-shot TTS to further work.
>
> - **"Comparison to SpeechTokenzier"**
>
>     Thanks for your advice. We will add a citation of SpeechTokenizer in our paper. However, we would like to point out that SpeechTokenizer and RepCodec focus on different categories of the discrete units. RepCodec aims to improve the quality of semantic units, trying to preserve more information for downstream tasks. On the other hand, SpeechTokenzier targets at improving the quality of acoustic units. Therefore, we do not think directly comparing numbers of these two methods is meaningful.

---

> > ### Comment · Reviewer_XznZ · 2023-11-23
> >
> > Thanks the authors for replying my comments and adding missing the reference, I keep my rating of the paper.

---

### Meta-Review · Area_Chair_jmxF · 2023-12-05

**Metareview:**

The paper modifies the autoencoder architecture proposed in EnCodec and SoundStream, by piping speech to HuBERT followed by a quantization layer.

The paper does not seem to be clear about its goal. On the surface, the goal seems to be looking for a representation that is accessible for ASR while being compact (having a low bit-rate) and complete (for re-synthesis). However, by this definition, EnCodec is already accessible for ASR while being compact and complete, albeit a bit inconvenient that we need to go back to wave form.

The above problem leads to Reviewer XznZ, wBo4, gJKh asking for more evaluation and Reviewer 6oit questioning the novelty. Reviewer wBo4's observation that this paper is not about learning a general representation for many tasks is spot-on. The authors should think more deeply about what the paper is about and how to position the paper.

The paper is also a victim of the term "semantic tokens" first introduced in AudioLM. It is a terribly unfortunate term, because HuBERT codes are more phonetic than phonemic, and are only somewhat lexical. HuBERT codes might contain prosodic information that is useful to differentiate meanings, but claiming that HuBERT codes are semantic (i.e., sufficient to extract meaning) is plain wrong. Being able to do ASR is also not sufficient to extract meaning.

The above problem leads to Review wBo4 asking for a more focused evaluation on language and Reviewer gJKh questioning the term "semantic quality".

The math is sloppy, but is relatively minor compared to the problems above.

Overall, the authors should think more carefully about what the goal really is for this work, what benefit would the approach bring, and having your own opinion on what convention to follow and what not to.

**Justification For Why Not Higher Score:**

The goal of the paper is unclear, and certain terms in the paper are misused, causing the readers to be confused.

**Justification For Why Not Lower Score:**

Rejection is the lowest score possible.

---

### Decision · Program_Chairs · 2024-01-16

Reject